Genomic diversity of Escherichia coli from healthy children in rural Gambia

Foster-Nyarko Ebenezer 1 2
Alikhan Nabil-Fareed 1
Ikumapayi Usman N. 2
Sarwar Golam 2
Okoi Catherine 2
Tientcheu Peggy-Estelle Maguiagueu 2
Defernez Marianne 1
O’Grady Justin 1
Antonio Martin 2 3
Pallen Mark J. m.pallen@warwick.ac.uk mark.pallen@quadram.ac.uk 1 4
1 Quadram Institute Bioscience, Norwich Research Park , Norfolk , United Kingdom
2 Medical Research Council Unit The Gambia at the London School of Hygiene and Tropical Medicine , Fajara , The Gambia
3 Microbiology and Infection Unit, Warwick Medical School, University of Warwick , Coventry , United Kingdom
4 School of Veterinary Medicine, University of Surrey , Surrey , United Kingdom
Souza Valeria
Electronic publication date: 2021 Jan 6
Publication date: 2021
Volume: 9
Electronic Location ID: e10572
Received 2020 Sep 8; Accepted 2020 Nov 23
Copyright: ©2021 Foster-Nyarko et al.
Copyright year: 2021
Copyright holder: Foster-Nyarko et al.
License: This is an open access article distributed under the terms of the Creative Commons Attribution License, which permits unrestricted use, distribution, reproduction and adaptation in any medium and for any purpose provided that it is properly attributed. For attribution, the original author(s), title, publication source (PeerJ) and either DOI or URL of the article must be cited.
License URL: https://creativecommons.org/licenses/by/4.0/

Keywords: Escherichia coli, Genomic diversity, Within-host evolution

Funding: Medical Research Council Unit, The Medical Research Council Unit The Gambia at London School of Hygiene and Tropical Medicine” is one name The Gambia at London School of Hygiene and Tropical Medicine BBSRC Institute Strategic Programme Microbes in the Food Chain (BB/R012504/1 and its constituent projects 44414000A and 4408000A) Core Capability Grant BB/ CCG1860/1 Martin Antonio, Usman N. Ikumapayi, Golam Sarwar, Catherine Okoi, Peggy-Estelle Maguiagueu Tientcheu and Mark J. Pallen were supported by the Medical Research Council Unit, The Gambia at London School of Hygiene and Tropical Medicine. The BBSRC Institute Strategic Programme, Microbes in the Food Chain (BB/R012504/1 and its constituent projects 44414000A and 4408000A) supported Ebenezer Foster-Nyarko and Mark J. Pallen. Marianne Defernez and Nabil-Fareed Alikhan were supported by the Quadram Institute Bioscience BBSRC funded Core Capability Grant (project number BB/ CCG1860/1). The funders had no role in study design, data collection and analysis, decision to publish, or preparation of the manuscript.

==============================
Little is known about the genomic diversity of Escherichia coli in healthy children from sub-Saharan Africa, even though this is pertinent to understanding bacterial evolution and ecology and their role in infection. We isolated and whole-genome sequenced up to five colonies of faecal E. coli from 66 asymptomatic children aged three-to-five years in rural Gambia (n = 88 isolates from 21 positive stools). We identified 56 genotypes, with an average of 2.7 genotypes per host. These were spread over 37 seven-allele sequence types and the E. coli phylogroups A, B1, B2, C, D, E, F and Escherichia cryptic clade I. Immigration events accounted for three-quarters of the diversity within our study population, while one-quarter of variants appeared to have arisen from within-host evolution. Several isolates encode putative virulence factors commonly found in Enteropathogenic and Enteroaggregative E. coli, and 53% of the isolates encode resistance to three or more classes of antimicrobials. Thus, resident E. coli in these children may constitute reservoirs of virulence- and resistance-associated genes. Moreover, several study strains were closely related to isolates that caused disease in humans or originated from livestock. Our results suggest that within-host evolution plays a minor role in the generation of diversity compared to independent immigration and the establishment of strains among our study population. Also, this study adds significantly to the number of commensal E. coli genomes, a group that has been traditionally underrepresented in the sequencing of this species.

Introduction

Ease of culture and genetic tractability account for the unparalleled status of Escherichia coli as “the biological rock star”, driving advances in biotechnology (Blount, 2015), while also providing critical insights into biology and evolution (Good et al., 2017). However, E. coli is also a widespread commensal, as well as a versatile pathogen, linked to diarrhoea (particularly in the under-fives), urinary tract infection, neonatal sepsis, bacteraemia and multi-drug resistant infection in hospitals (Camins et al., 2011; Rodríguez-Baño et al., 2010; Russo & Johnson, 2003). Yet, most of what we know about E. coli stems from the investigation of laboratory strains, which fail to capture the ecology and evolution of this key organism “in the wild” (Hobman, Penn & Pallen, 2007). What is more, most studies of non-lab strains have focused on pathogenic strains or have been hampered by low-resolution PCR methods, so we have relatively few genomic sequences from commensal isolates, particularly from low- to middle-income countries (Ahmed et al., 2014; Ferjani et al., 2017; Moremi et al., 2017; Oshima et al., 2008; Rasko et al., 2008; Stoesser et al., 2015; Touchon et al., 2009).

We have a broad understanding of the population structure of E. coli, with eight significant phylogroups loosely linked to ecological niche and pathogenic potential (B2, D and F linked to extraintestinal infection; A and B1 linked to severe intestinal infections such as haemolytic-uraemic syndrome) (Alm, Walk & Gordon, 2011; Escobar-Paramo et al., 2004a; Escobar-Páramo et al., 2004b; Mellata, 2013; Walk et al., 2009). All phylogroups can colonise the human gut, but it remains unclear how far commensals and pathogenic strains compete or collaborate—or engage in horizontal gene transfer—within this important niche (Laxminarayan et al., 2013; Stoppe et al., 2017).

Although clinical microbiology typically relies on single-colony picks (which has the potential to underestimate species diversity and transmission events), within-host diversity of E. coli in the gut is crucial to our understanding of inter-strain competition and co-operation and also for accurate diagnosis and epidemiological analyses. Pioneering efforts using serotyping, molecular typing and whole-genome sequencing have shown that normal individuals typically harbour more than one strain of E. coli, with one individual carrying 24 distinct clones (Chen et al., 2013; Schlager et al., 2002; Shooter et al., 1977; Dixit et al., 2018; Richter et al., 2018; Bettelheim, Faiers & Shooter, 1972; Sears, Brownlee & Uchiyama, 1950; Sears & Brownlee, 1952). More recently, whole-genome sequencing has illuminated molecular epidemiological investigations (Stoesser et al., 2015), for example, studies of the transmission of extended-spectrum beta-lactamase-encoding E. coli, multidrug-resistant Acinetobacter baumannii, and the genomic surveillance of multidrug-resistant E. coli carriage. Whole-genome data has also been applied to studies of E. coli adaptation during and after infection (McNally et al., 2013; Nielsen et al., 2016), as well as the intra-clonal diversity in healthy hosts (Stegger et al., 2020).

There are two plausible sources of within-host genomic diversity. Although a predominant strain usually colonises the host for extended periods (Hartl & Dykhuizen, 1984), successful immigration events mean that incoming strains can replace the dominant strain or co-exist alongside it as minority populations (Bettelheim, Faiers & Shooter, 1972). Strains originating from serial immigration events are likely to differ by hundreds or thousands of single-nucleotide polymorphisms (SNPs). Alternatively, within-host evolution can generate clouds of intra-clonal diversity, where genotypes differ by just a handful of SNPs (Dixit et al., 2018).

Most relevant studies have been limited to Western countries, except for a recent report from Tanzania (Richter et al., 2018), so, little is known about the genomic diversity of E. coli in sub-Saharan Africa. The Global Enteric Multicenter Study (GEMS) (Kotloff et al., 2013; Liu et al., 2016) has documented a high burden of diarrhoea attributable to E. coli (including Shigella) among children from the Gambia, probably as a result of increased exposure to this organism through poor hygiene and frequent contact with animals and the environment. GEMS was a prospective case-control study which investigated the aetiology of moderate-to-severe diarrhoea in children aged less than five years residing in sub-Saharan Africa and South Asia. In the Gambia, children with moderate-to-severe diarrhoea seeking care at the Basse Health centre in the Upper River Division of the country were recruited, with one to three matched control children randomly selected from the community along with each case. In also facilitating access to stool samples from healthy Gambian children, the GEMS study has given us a unique opportunity to study within-host genomic diversity of commensal E. coli in this setting.

Methods

Study population

We initially selected 76 faecal samples from three- to five-four-old (36–59 months) asymptomatic Gambian children, who had been recruited into the GEMS study (Kotloff et al., 2013) as healthy controls from December 1, 2007, to March 3, 2011. Samples had been collected according to a previously described sampling protocol (Kotloff et al., 2012) and the results of the original study are publicly available at ClinEpiDB.org. Ten of the original 76 samples were depleted and were therefore unavailable for processing in this study. Of the remaining 66 stools, 62 had previously tested positive for E. coli. GEMS isolated three E. coli colonies per stool sample but pooled these into a single tube for frozen storage. Thus, we needed to re-culture the stools with multiple colony picks, as the original isolate collection was unsuitable for the investigation of within-host diversity. Archived stool samples were retrieved from −80 °C storage and allowed to thaw on ice. A 100–200 mg aliquot from each sample was transferred aseptically into 1.8 ml Nunc tubes for microbiological processing below (Fig. 1).

Figure 1 The study sample-processing flow diagram.

Bacterial growth and isolation

1 ml of physiological saline (0.85%) was added to each sample tube and vigorously vortexed at 4,200 rpm for at least 2 min. Next, the homogenised sample suspensions were taken through four ten-fold dilution series. A100 µl aliquot from each dilution was then spread evenly on a plate of tryptone-bile-X-glucuronide differential and selective agar. The inoculated plates were incubated overnight at 37 °C under aerobic conditions. Colony counts were performed on the overnight cultures for each serial dilution for translucent colonies with entire margins and blue–green pigmentation indicative of E. coli. Up to five representative colonies were selected from each sample and sub-cultured on MacConkey agar overnight at 37 °C before storing in 20% glycerol broth at −80 °C. Individual isolates were assigned a designation comprised of the subject ID followed by the colony number (“1–5”).

Genomic DNA extraction and genome sequencing

Broth cultures were prepared from pure, fresh cultures of each colony-pick in 1 ml Luria-Bertani broth and incubated overnight to attain between 109–1010 cfu per ml. Genomic DNA was then extracted from the overnight broth cultures using the lysate method described in Foster-Nyarko et al. (2020). The eluted DNA was quantified by the Qubit high sensitivity DNA assay kit (Invitrogen, MA, USA) and sequenced on the Illumina NextSeq 500 instrument (Illumina, San Diego, CA), using a modified Nextera XT DNA protocol for the library preparation as described previously (Foster-Nyarko et al., 2020). The pooled library was loaded on a mid-output flow cell (NSQ 500 Mid Output KT v2 300 cycles; Illumina Catalogue No. FC-404–2003) at a final concentration of 1.8 pM, following the Illumina recommended denaturation and loading parameters—including a 1% PhiX spike (PhiX Control v3; Illumina Catalogue FC-110–3001).

Following Dixit et al. (2018), we sequenced a random selection of ten isolates twice, using DNA obtained from independent cultures, to help in the determination of clones and the analysis of within-host variants (File S1). Bioinformatic analyses of the genome sequences were carried out on the Cloud Infrastructure for Microbial Bioinformatics (CLIMB) platform (Connor et al., 2016).

Genome assembly and phylogenetic analysis

The paired 150 bp reads were concatenated, then quality checked using the FastQC tool v0.11.7 (Wingett & Andrews, 2018) and assembled using SPAdes genome assembler v3.12.0 (Bankevich et al., 2012), under default parameters. The quality of the assemblies was assessed using QUAST v5.0.0, de6973bb (Gurevich et al., 2013). We used Snippy v4.3.2 (https://github.com/tseemann/snippy)—a rapid command line tool that finds SNPs (substitutions and insertions/deletions) between a haploid reference genome and input sequence reads and generates a core SNP alignment which can be used to reconstruct a high-resolution phylogeny—to generate a core-genome alignment based on core SNPs under default parameters. The complete genome sequence of commensal E. coli str. K12 substr. MG1655 was used as a reference strain (NCBI accession: NC_000913.3). From the core-genome alignment, we then reconstructed a maximum-likelihood phylogeny with 1,000 bootstrap replicates using RAxML v8.2.4 (Stamatakis, 2006), based on a general time-reversible nucleotide substitution model. The phylogenetic tree was rooted using the genomic sequence of E. fergusonii as an outgroup (NCBI accession: GCA_000026225.1). The phylogenetic tree was visualised in FigTree v1.4.3 (https://github.com/rambaut/figtree/) and annotated in RStudio v3.5.1 and Adobe Illustrator v 23.0.3 (Adobe Inc., San Jose, California). As recombination is known to be widespread in E. coli and can blur phylogenetic signals (Wirth et al., 2006), we detected and masked any recombinant regions of the core-genome alignment using Gubbins (Genealogies Unbiased By recomBinations In Nucleotide Sequences) (Croucher et al., 2015) before the phylogenetic reconstruction. For visualisation, a single colony was chosen to represent replicate colonies of the same strain (ST) with identical virulence, plasmid and antimicrobial resistance profiles and a de-replicated phylogenetic tree reconstructed using the representative isolates. We computed pairwise SNP distances between genomes from the core-genome alignment using snp-dists v0.6 (https://github.com/tseemann/snp-dists).

Multi-locus sequence typing and Clermont typing

The merged reads were uploaded to EnteroBase (Zhou et al., 2020), where de novo assembly and genome annotation were carried out, and in-silico multi-locus sequence types (MLST) assigned based on the Achtman scheme, allocating new sequence types (ST) if necessary. EnteroBase assigns phylogroups using ClermontTyper and EzClermont (Clermont et al., 2013; Clermont, Gordon & Denamur, 2015) and unique core-genome MLST types (cgMLST) based on 2, 513 core loci in E. coli. Publicly available E. coli sequences in EnteroBase (http://enterobase.warwick.ac.uk/species/index/ecoli) (Zhou et al., 2020) were included for comparative analysis, including 23 previously sequenced isolates obtained from diarrhoeal cases recruited in the GEMS study in the Gambia (File S2). The isolates can be searched in EnteroBase using the ‘Search Strains’ parameter and under ‘Strain Metadata’, selecting the ‘Name’ option and entering the study sample name (column 1 of File S2) in the ‘Value’ box.

Determination of immigration events and within-host variants

For the whole genome sequences of the strains sequenced twice, we used SPAdes v3.13.2 (Bankevich et al., 2012) to assemble each set of reads and map the raw sequences from one sequencing run to the assembly of the other run and vice versa, as described previously (Dixit et al., 2018). Briefly, mapping was done using the BWA-MEM algorithm v0.7.17-r1188 under default parameters to generate a SAM alignment. This was then converted to BAM files using Samtools view v1.9 (Li et al., 2009), sorted and indexed. Next, variants were called and written to a VCF file using Samtools mpileup and the “view” module of BCFtools (which is part of the Samtools v1.9 package) and visualised in Tablet v1.19.09.13 (Milne et al., 2013). The number of SNPs, and their positions were determined and compared between the two steps, counting only those SNPs that were detected in both sets of reads as accurate.

In line with Dixit et al. (2018), isolates belonging to different STs recovered from the same host were considered to be separate strains derived from independent exposures and immigration events. As described in Dixit et al. (2018), we determined the number of SNP differences that existed between assemblies of the same isolate that were sequenced on two separate occasions, to determine if multiple isolates of the same ST from a single host were distinct variants (clones). If the SNP difference between two isolates belonging to the same ST recovered from the same host was less than the SNP difference between the sequences of the same isolate sequenced on two separate occasions, then the two isolates were taken to represent replicate copies of the same clone. Otherwise, they were considered as within-host variants (separate, distinct clones of the same strain)—provided the SNP differences between such distinct clones were no more than eleven SNPs. This cut-off was chosen based on an estimated mutation rate of 1.1 SNP per genome per year (Reeves et al., 2011), assuming equal rates of mutation in both genomes being compared. Based on these data, we inferred replicate clones with SNP differences of greater than 11 SNPs to represent a divergence of more than five years. Thus, it seems implausible that such replicate clones would have emerged from within-host evolution, considering the age of the study participants (<5 years old).

We produced a contingency table to summarise the distribution of variants derived from migration events and within-host evolution and visualised this using a clustered bar graph. We then performed Fisher’s exact test to investigate the association between phylogroup and the distribution of variants (migration versus within-host evolution). Our calculations were based on the assumption of independence among the observed phylogroups—that is, the finding of one phylogroup does not preclude or predict the co-occurrence of another.

Accessory gene content

We used ABRicate v0.9.8 (https://github.com/tseemann/abricate) to predict virulence factors, acquired antimicrobial resistance (AMR) genes and plasmid replicons by scanning the contigs against the VFDB, ResFinder and PlasmidFinder databases respectively, using an identity threshold of ≥ 90% and a coverage of ≥ 70%. Virulence factors and AMR genes were plotted next to the phylogenetic tree using the ggtree, ggplot2 and phangorn packages in RStudio v3.5.1. We calculated co-occurrence of AMR genes among study isolates by transforming the binary AMR gene content matrix and visualising this as a heat map using the pheatmap package v 1.0.12 (https://CRAN.R-project.org/package=pheatmap) in RStudio v3.5.1. We computed Fisher’s exact tests between the detected virulence factors and the observed phylogroups in RStudio v3.5.1.

Population structure and comparison of commensal and pathogenic strains

We assessed the population structure using the hierarchical clustering algorithm in EnteroBase. Briefly, the isolates were assigned stable population clusters at eleven levels (from HC0 to HC 2350) based on pairwise cgMLST allelic differences. Hierarchical clustering at 1,100 alleles differences (HC1100) resolves populations into cgST (core-genome MLST type) complexes, the equivalent of clonal complexes achieved with the legacy MLST clustering approaches (Zhou et al., 2020). We reconstructed neighbour-joining phylogenetic trees using NINJA (Wheeler, 2009), based on clustering at HC1100 to display the population sub-clusters at this level as an indicator of the genomic diversity within our study population and to infer the evolutionary relationship among our strains and others in the public domain.

Next, we interrogated the HC1100 clusters that encompassed our study isolates and Gambian pathogenic isolates recovered from diarrhoeal cases and commensal E. coli strains recovered from the GEMS study. For the clusters that encompassed commensal and pathogenic strains belonging to the same ST (HC1100_200 cluster, comprising pathogenic isolates from GEMS cases 100415, 102106 and 102098 and the resident ST38 strain recovered from our study subject 18), we reconstructed both neighbour-joining and SNP phylogenetic trees to display the genetic relationships among these strains. We visualised the accessory genomes for the overlapping STs mentioned above to determine genes associated with phages, virulence factors and AMR. The resulting phylogenetic trees were annotated in Adobe Illustrator v 23.0.3 (Adobe Inc., San Jose, California).

Ethical statement

The parent study was approved by the joint Medical Research Council Unit The Gambia-Gambian Government ethical review board (SCC 1331). Written informed consents were obtained from all the study participants as previously reported in Kotloff et al. (2013). The Medical Research Council Unit The Gambia at London School of Hygiene and Tropical Medicine’s Scientific Coordinating Committee gave approval for the use of the stool samples analysed in this study.

Results

Population structure

The study population included 27 females and 39 males (File S3). All but one reported the presence of a domestic animal within the household. Twenty-one samples proved positive for the growth of E. coli, yielding 88 isolates (File S4). We detected 37 seven-allele sequence types (STs) among the isolates, with a fairly even distribution (Fig. 2). Five STs were completely novel (ST9274, ST9277, ST9278, ST9279 and ST9281). These study strains were scattered over all the eight main phylogroups of E. coli: A (27%), B1 (32%), B2 (9%), D (15%), C and F (5% each), E (1%), and the cryptic Clade I (7%), although the majority belonged to phylogroups A and B1 (Table 1). Hierarchical clustering of core genomic STs revealed twenty-seven cgST clonal complexes (File S4). The raw genomic sequences of the study isolates have been deposited in the NCBI SRA under the BioProject ID PRJNA658685, (accession numbers SAMN15880274 to SAMN15880361).

Figure 2 A maximum-likelihood tree depicting the phylogenetic relationships among the study isolates.

The tree was reconstructed with RAxML, using a general time-reversible nucleotide substitution model and 1,000 bootstrap replicates. The genome assembly of E. coli str. K12 substr. MG1655 was used as the reference, and the tree rooted using the genomic assembly of E. fergusonii as an outgroup. The sample names are indicated at the tip, with the respective Achtman sequence types (ST) indicated beside the sample names. The respective phylogroups the isolates belong to are indicated with colour codes as displayed in the legend. The E. coli reference genome and E. fergusonii are denoted in black. Asterisks (*) are used to indicate novel STs. The predicted antimicrobial resistance genes and putative virulence factors for each isolate are displayed next to the tree, with the virulence genes clustered according to their function. Multiple copies of the same strain (ST) isolated from a single host are not shown. Instead, we have shown only one representative isolate from each strain. Virulence and resistance factors were not assessed in the reference strains either. A summary of the identified virulence factors and their known functions are provided in File S3.

Table 1 Phylogroup and sequence types of the distinct clones isolated in each study subject.

	Colony or isolate number	Number of distinct genotypes (clones)	Migration events	Within-host evolution events	
Host	1	2	3	4	5		Phylotype (number of events)	Phylotype (number of events)	
H-2	A (9274)	A (9274)	A (9274)	A (9274)	A (9274)	1	A (1)	0	
H-9	A (2705)	A (2705)	A (2705)	D (2914)	B1 (29)	3	A (1), D (1), B1 (1)	0	
H-15	B2 (9277)	B2 (9277)	B2 (9277)	Clade I (747)	Clade I (747)	3	B2 (1), Clade I (1)	Clade I (1)	
H-18	D (38)	D (38)	B1 (9281)	A (9274)		4	D (1), B1 (1), A (1)	D (1)	
H-21	B1 (58)	B1 (58)	B1 (223)	A (540)	D (1204)	4	B1(2) A (1), D (1)	0	
H-22	B1 (316)	B1 (316)	B1 (316)	B1 (316)		2	B (1)	B1(1)	
H-25	A (181)	A (181)	A (181)	A (181)	B1 (337)	4	A (1), B1 (1)	A (2)	
H-26	B1 (641)	B1 (2741)	A (10)	A (398)		4	B1(2), A (1), D (1)	0	
H-28	B1 (469)	B1 (469)	B1 (469)	B1 (469)		2	B1(1)	B1(1)	
H-32	B1 (101)	B1 (101)	B1 (101)	B1 (2175)	A (10)	3	B1(2), A (1)	0	
H-34	B1 (603)	B1 (603)	B1 (603)	B1 (1727)	A (10)	4	B1(2), A (1)	B1(1)	
H-35	A (226)					1	A (1)	0	
H-36	F (59)	F (59)	F (59)	F (59)	E (9278)	4	F (1), E (1)	F (1)	
H-37	D (5148)	D (5148)	D (5148)	D (5148)	D (5148)	3	D (1)	D (2)	
H-38	D (394)	D (394)	D (394)	D (394)	B1 (58)	4	D (1), B1(1)	D (2)	
H-39	B2 (452)	B2 (452)	B2 (452)	B2 (452)	B2 (452)	2	B2(1)	B2 (1)	
H-40	B1 (155)					1	B1(1)	0	
H-41	A (43)	A (43)	A (43)	A (43)	B1 (9283)	2	A (1), B1(1)	0	
H-48	Clade I (485)	Clade I (485)	Clade I (485)	Clade I (485)		3	Clade I (1)	0	
H-50	C (410)	C (410)	C (410)	C (410)	B1 (515)	2	C (1), B1(1)	0	
H-55	A (9279)					1	A(1)	0	

Within-host diversity

Just a single ST colonised nine individuals, six carried two STs, four carried four STs, and two carried six STs. We found 56 distinct genotypes, which equates to an average of 2.7 genotypes per host. Two individuals (H-18 and H-2) shared an identical strain belonging to ST9274 (zero SNP difference) (File S5, yellow highlight), suggesting recent transfer from one child to another or recent acquisition from a common source.

We observed thirteen within-host variants in ten hosts (intra-clonal diversity) (subjects H-15, H-18, H-22, H-25, H-28, H-34, H-36, H-37, H-38 and H-39), compared to forty-one immigration events (Tables 1 and 2). Overall, immigration events accounted for the majority (76%) of variants (Fig. S1). The proportion of migration versus within-host evolution events did not appear to be affected by phylogroup (p = 0.42). Twenty-two percent of within-host mutations represented synonymous changes, 43% were non-synonymous mutations, while 31% occurred in non-coding regions, and 4% represented stop-gained mutations (File S6). On an average, Ka/Ks ratios were greater than 1, which seems to suggest that these mutations were under positive Darwinian selection—indicating that most of the mutations were likely to have little effect on fitness. However, these remain to be investigated further. Also, the observed non-synonymous mutations were spread across genes with a variety of functions, including metabolism, transmembrane transport, pathogenesis and iron import into the cell. However, the bulk (42%) occurred in genes involved in metabolism. The average number of SNPs among within-host variants was 5 (range 0–18) (Table 2). However, in two subjects (H36 and H37), pairwise distances between genomes from the same ST (ST59 and ST5148) were as large as 14 and 18 SNPs respectively (File S5, grey highlight).

Table 2 Pairwise SNP distances between variants arising from within-host evolution.

Host	Sequence type (ST)	Colonies per ST	Pairwise SNP distances between multiple colonies of the same ST	
H2	9274	5	0–9	
H9	2705	3	0–1	
H15	9277	3	0–1	
H15	747	2	3	
H18	38	2	3	
H21	58	2	0	
H22	316	4	0–3	
H25	181	4	1–5	
H28	469	4	0–3	
H32	101	3	1–9	
H34	603	3	2–8	
H36	59	4	0–14	
H37	5148	5	2–18	
H38	394	4	1–3	
H39	452	5	0–2	
H41	43	4	0–1	
H48	485	4	1–9	
H50	410	4	0	

Figure 3 The population structure of ST38.

(A) A NINJA neighbour-joining tree showing the population structure of E. coli ST38, drawn using the genomes found in the core-genome MLST hierarchical cluster at HC1100, which corresponds to ST38 clonal complex. The size of the nodes represents the number of isolates per clade. The geographical locations where isolates were recovered are displayed in the legend; with the genome counts shown in square brackets. The study resident ST38 strains and the pathogenic ST38 strains recovered from GEMS cases are highlighted with red circles around the nodes. (B) The closest neighbour to a pathogenic strain reported in GEMS Kotloff et al. (2013) is shown to be a commensal isolate recovered from a healthy individual. The size of the nodes represents the number of isolates per clade. The geographical locations where isolates were recovered are displayed in the legend; with the genome counts shown in square brackets. Red circles around the nodes are used to highlight the study resident ST38 strains and the pathogenic ST38 strains recovered from GEMS cases within this cluster. (C) The closest relatives to the commensal ST38 strain recovered from this study is shown (red highlights), with the number of core-genome MLST alleles separating the two genomes displayed. The geographical locations where isolates were recovered are displayed in the legend; with the genome counts shown in square brackets, and the size of the nodes depicting the number of isolates per clade. (D) A maximum-likelihood phylogenetic tree reconstructed using the genomes found in the cluster in C above, comprising both pathogenic and commensal ST38 strains is presented, depicting the genetic relationship between strain 100415 (pathogenic) and 103709 (commensal) (red highlights). The nodes are coloured to depict the status of the strains, as pathogenic (red) or commensal (blue). The size of the nodes represents the number of isolates per clade. The geographical locations where isolates were recovered are displayed in the legend; with the genome counts shown in square brackets.

Accessory gene content and relationships with other strains

A quarter of our isolates were most closely related to commensal strains from humans, with smaller numbers most closely related to human pathogenic strains or strains from livestock, poultry or the environment (File S7). One isolate was most closely related to a canine isolate from the UK. Three STs (ST38, ST10 and ST58) were shared by our study isolates and diarrhoeal isolate from the GEMS study (Fig. S2), with just eight alleles separating our commensal ST38 strain from a diarrhoeal isolate from the GEMS study (Fig. 3). For ST10 and ST58, hierarchical clustering placed the commensal strains from this study into separate clusters from the pathogenic isolates from diarrhoeal cases, indicating that they were genetically distinct to each other. Yet, the closest relative of our study ST58 strain was an extraintestinal strain isolated from the blood of a 69-year-old male (87 alleles differences, Fig. 4). Also, the resident ST10 isolates recovered from this study (H-26_2, H-34_2, and H-32_5) had their closest neighbours in isolates from livestock (83 and 111 alleles each), and an isolate of an unspecified source (18 alleles differences) respectively (File S7).

Figure 4 The population structure of ST58.

(A) A NINJA neighbour-joining tree depicting the population structure of E. coli ST58, drawn using the genomes found that clustered together in the same HC1100 hierarchical cluster in the core-genome MLST scheme in EnteroBase (Zhou et al., 2020). Commensal ST58 strains from this study and Gambian pathogenic ST58 isolates from GEMS are highlighted in red. The geographical locations where isolates were recovered are displayed in the legend; with the genome counts shown in square brackets. The size of the nodes represents the number of isolates per clade. (B and C) The closest relatives to the study ST58 strains are shown. Geographical locations where isolates were recovered are displayed in the legend, with the genome counts displayed in square brackets. The red highlights around the nodes depict the study commensal ST58 strains and their closest neighbours. The size of the nodes represents the number of isolates per clade, and the geographical locations where isolates were recovered are displayed in the legend; with the genome counts shown in square brackets.

We detected 130 genes encoding putative virulence factors across the 88 study isolates (Fig. 2; File S8). Notable among these were genes associated with pathogenesis in Enteroaggregative E. coli and Salmonella referred to as the Serine Protease Autotransporters of Enterobacteriaceae (SPATEs) (Pokharel et al., 2019), such as sat (13%), sigA (11%) and pic (1%). Besides, eight isolates harboured known markers of Enteropathogenic E. coli (eltAB or estA). Several strains (across all phylogroups) also harboured virulence genes associated with intestinal or extraintestinal disease in humans, including adhesins, invasins, toxins and iron-acquisition genes such as fyuA, several fim and pap genes, iroN, irp1, 2, ibeA and aslA. We did not detect any of the well-known markers of EPEC (eae, bfpA, stx1, or stx2) (Fig. 2, File S8).

The prevalence of some virulence factors involved in invasion/evasion, iron uptake, adherence, and secretion systems appeared to be more or less likely to occur in one or a few phylotypes (p ≤ 0.05) as follows (File S9). The iron acquisition genes chuA, S-Y and shuA, S, T, Y were found to be present in all cases for phylogroup D (n = 5), and absent in virtually all cases for phylogroups A (n = 13) and B1 (n = 16). On the other hand, iutA and iucA-D were observed in the two cases from phylogroup B2, and absent from all samples from phylogroup D (n = 5). The invasion/evasion genes kpsD, M, T and aslA were found to be present in almost all cases for phylogroups D (n = 5), B2 (n = 2), and Clade I (n = 2), and absent in B1 (n = 16). The secretion system gene cluster espB, D, G, K-N, R, W-Y was observed in all cases except the two belonging to phylogenetic group B2. The protease gene sigA was absent from most samples, except two samples from phylotype B2. The adherence gene fdeC was observed in all cases for phylotype D (n = 5) and most for B1 (n = 16).

More than half of the isolates encoded resistance to three or more clinically relevant classes of antibiotics such as aminoglycosides, penicillins, trimethoprim, sulphonamides and tetracyclines (Fig. 5; Fig. S3). The most common resistance gene network was -aph(6)-Id_1-sul2 (41% of the isolates), followed by aph(3″)-Ib_5-sul2 (27%) and bla-TEM-aph(3″)-Ib_5 (24%). Most isolates (67%) harboured two or more plasmid types (Fig. 6). Of the 24 plasmid types detected, IncFIB was the most common (41%), followed by col156 (19%) and IncI_1-Alpha (15%). Nearly three-quarters of the multi-drug resistant isolates carried IncFIB (AP001918) plasmids (∼50 kb), suggesting that these large plasmids may be linked to the dissemination of resistance genes within our study population.

Figure 5 The prevalence of antimicrobial-associated genes detected in the study isolates.

(A) The y-axis shows the prevalence of the detected AMR-associated genes in the study isolates, grouped by antimicrobial class. (B) A histogram depicting the number of antimicrobial classes to which resistance genes were detected in the corresponding strains.

Figure 6 Prevalence of plasmid replicons among the study isolates.

(A) Plasmid replicons detected in the study isolates. (B) A histogram depicting the number of plasmids co-harboured in a single strain.

Discussion

This study provides an overview of the within-host genomic diversity of E. coli in healthy children from a rural setting in the Gambia, West Africa. Surprisingly, we were able to recover E. coli from only 34% of stools which had previously tested positive for E. coli in the original study. This low rate of recovery may reflect some hard-to-identify effect of long-term storage (nine to thirteen years) or the way the samples were handled, even though they were kept frozen and thawed only just before culture.

Several studies have shown that sampling a single colony is insufficient to capture E. coli strain diversity in stools (Dixit et al., 2018; Richter et al., 2018; Shooter et al., 1977). Lidin-Janson et al. (1978) claim that sampling five colonies provides a >99% chance of recovering dominant genotypes from single stool specimens, while Schlager et al. (2002) calculate that sampling twenty-eight colonies provides a >90% chance of recovering minor genotypes. Our results confirm the importance of multiple-colony picks in faecal surveillance studies, as over half (57%) of our strains would have been missed by picking a single colony.

Our strains encompassed all eight major phylotypes of E. coli, however, the majority fell into the A and B1 phylogenetic groups, in line with previous reports that these phylogroups dominate in stools from people in low- and middle-income countries (Duriez et al., 2001; Escobar-Paramo et al., 2004a; Escobar-Páramo et al., 2004b). Although not fully understood, there appear to be host-related factors that influence the composition of E. coli phylogroups in human hosts. For example, the establishment of strains belonging to phylogroups E or F seems to favour subsequent colonisation by other phylotypes, compared to the establishment of phylogroup B2 strains, which tend to limit the heterogeneity within individual hosts (Gordon, O’Brien & Pavli, 2015). Geographical differences have also been reported, with phylogroups A and B1 frequently dominating the stools of people living in developing countries (Duriez et al., 2001; Escobar-Paramo et al., 2004a; Escobar-Páramo et al., 2004b). Conversely, phylogroups B2 and D strains appear to be pervasive among people living in developed countries (Massot et al., 2016; Skurnik et al., 2008). These locale-specific patterns in the distribution of E. coli phylotypes have been attributed to differences in diet and climate (Duriez et al., 2001; Escobar-Paramo et al., 2004a; Escobar-Páramo et al., 2004b).

The prevalence of putative virulence genes in most of our isolates highlights the pathogenic potential of commensal intestinal strains—regardless of their phylogroup—should they gain access to the appropriate tissues, for example, the urinary tract. Our results complement previous studies reporting genomic similarities between faecal E. coli isolates and those recovered from urinary tract infection (McNally et al., 2013; Wold et al., 1992).

We found that within-host evolution plays a minor role in the generation of diversity in our study population. This might be due to the low prevalence of B2 strains, which are thought to inhibit the establishment of strains from other phylogroups, as discussed above (Gordon, O’Brien & Pavli, 2015); or it may indicate that members of phylogroups A and B1 might favour a more heterogeneous composition of E. coli phylotypes in stools of healthy individuals. However, this remains to be properly investigated, as we did not find statistical evidence that the distribution of variants (independent migration versus within-host evolution) was influenced by phylogroup. Our findings are similar to that reported by Dixit et al. (2018), who reported that 83% of diversity originates from immigration events, and with epidemiological data suggesting that the recurrent immigration events account for the high faecal diversity of E. coli in the tropics (Tenaillon et al., 2010).

The estimated mutation rate for E. coli lineages is around one SNP per genome per year (Reeves et al., 2011), so that two genomes with a most recent common ancestor in the last five years would be expected to be around ten SNPs apart. However, in two subjects, pairwise distances between genomes from the same ST (ST59 and ST5148) were large enough (14 and 18 respectively) to suggest that they might have arisen from independent immigration events, as insufficient time had elapsed in the child’s life for such divergence to occur within the host. However, it remains possible that the mutation rate was higher than expected in these lineages, although we found no evidence of damage to DNA repair genes. Co-colonising variants belonging to the same ST tended to share an identical virulence, AMR and plasmid profile, signalling similarities in their accessory gene content.

The sources of novel variation that account for within-host diversity include point mutation and small insertions or deletions (indels), indels and the loss or acquisition of mobile genetic elements. Among the variants inferred to have been derived from within-host evolution, we observed a dominance of mutations that were predicted to result in changes in protein function, in the form of missense mutations and non-sense mutations (leading to a premature stop codon). Although the mutations appeared to be heterogeneously distributed, a higher number was observed in genes associated with metabolism. These appeared to be under positive selection, although it remains to be seen if these changes confer any effects on fitness. It will be desirable to investigate this in future studies. Due to the cross-sectional nature of our sampling, we were unable to analyse the dynamics of strain gain or loss and variation in gene content over time. Homologous recombination has also been noted to contribute to the generation of diversity (Golubchik et al., 2013; González-González et al., 2013), however, we detected and removed recombinant regions prior to phylogenetic reconstruction and thus focused our analysis on SNPs.

More than half of our isolates encode resistance to three or more classes of antimicrobials echoing the high rate of MDR (65%; confirmed by phenotypic testing) in the GEMS study. IncFIB (AP001918) was the most common plasmid Inc type from our study, in line with the observation that IncF plasmids are frequently associated with the dissemination of resistance (Carattoli, 2009). However, a limitation of our study is that we did not perform phenotypic antimicrobial resistance testing, although Doyle et al. (2020) reported that only a small proportion of genotypic AMR predictions are discordant with phenotypic results.

Comparative analyses confirm the heterogeneous origins of the strains reported here, documenting links to other human commensal strains or isolates sourced from livestock or the environment. This is not surprising, as almost all the study participants reported that animals are kept in their homes and children in rural Gambia are often left to play on the ground, close to domestic animals such as pets and poultry (Dione et al., 2011).

Conclusions

Our results show that the commensal E. coli population in the gut of healthy children in rural Gambia is richly diverse, with the independent immigration and establishment of strains contributing to the bulk of the observed diversity. An obvious limitation to our study is the low recovery of E. coli from frozen stools—which potentially implies we may have underestimated the extent of genetic diversity present within our study population. Although solely observational, our study paves the way for future studies aimed at a mechanistic understanding of the factors driving the diversification of E. coli in the human gut and what it takes to make a strain of E. coli successful in this habitat. Besides, this work has added significantly to the number of commensal E. coli genomes, which are underrepresented in public repositories.

Supplemental Information

Supplemental Information 1 List of the sample clones for which two independent cultures were obtained and sequenced to find the SNPs between the same clones

Click here for additional data file.

Supplemental Information 2 Sequencing statistics and characteristics of twenty-four previously sequenced GEMS cases included in this study

Click here for additional data file.

Supplemental Information 3 Characteristics of the study population

Click here for additional data file.

Supplemental Information 4 A summary of the sequencing statistics of the study isolates reported in this study

Click here for additional data file.

Supplemental Information 5 A pairwise single nucleotide polymorphism matrix showing the SNP distances between the study genomes

Click here for additional data file.

Supplemental Information 6 Mutations in variants inferred to have been derived from within-host evolution

Click here for additional data file.

Supplemental Information 7 Closest relatives to the study isolates

Click here for additional data file.

Supplemental Information 8 A summary of the virulence factors detected among the study isolates and their known functions

Click here for additional data file.

Supplemental Information 9 Fisher’.s exact tests of associations between phylogroups and the detected virulence factors

Click here for additional data file.

Supplemental Information 10 Variants derived from immigration events vs. within-host evolution

The distribution of variants inferred to have arisen from immigration events compared to those generated by within-host evolution by phylogroup

Click here for additional data file.

Supplemental Information 11 A Neighbour-joining phylogenetic tree depicting the genetic relationships among twenty-four strains isolated from diarrhoeal cases in the GEMS study (Kotloff et al. (2013); Liu et al. (2016))

The Sequence types identified in these isolates are shown in the legend, with the genome count displayed in square brackets next to the respective sequence types. Three STs (ST38, ST58 and ST10) overlapped with what was found among commensal strains from this study (see Fig. 2).

Click here for additional data file.

Supplemental Information 12 A co-occurrence matrix of acquired antimicrobial resistance genes detected in the study isolates

The diagonal values show how many isolates each individual gene was found in, while the intersections between the columns represent the number of isolates in which the corresponding antimicrobial resistance genes co-occurred

Click here for additional data file.

We gratefully acknowledge the study participants in GEMS and all clinicians, field workers and the laboratory staff of the Medical Research Council Unit The Gambia at London School of Hygiene and Tropical Medicine involved in the collection and storage of stools in the GEMS study in Basse Field Station and Fajara.

Additional Information and Declarations

Competing Interests

Author Contributions

Human Ethics

DNA Deposition

The authors declare there are no competing interests.

Ebenezer Foster-Nyarko conceived and designed the experiments, performed the experiments, analyzed the data, prepared figures and/or tables, authored or reviewed drafts of the paper, and approved the final draft.

Nabil-Fareed Alikhan, Usman N. Ikumapayi, Golam Sarwar, Catherine Okoi, Peggy-Estelle Maguiagueu Tientcheu, Marianne Defernez and Justin O’Grady performed the experiments, authored or reviewed drafts of the paper, and approved the final draft.

Martin Antonio conceived and designed the experiments, performed the experiments, authored or reviewed drafts of the paper, and approved the final draft.

Mark J. Pallen conceived and designed the experiments, performed the experiments, analyzed the data, authored or reviewed drafts of the paper, and approved the final draft.

The following information was supplied relating to ethical approvals (i.e., approving body and any reference numbers):

The study was approved by the Medical Research Council Unit The Gambia at London School of Hygiene and Tropical Medicine’s Scientific Coordinating Committee.

The following information was supplied regarding the deposition of DNA sequences:

All genomic assemblies for the strains included in this study are freely available from EnteroBase (http://enterobase.warwick.ac.uk/species/index/ecoli). The EnteroBase genome assembly barcodes are available in the Supplemental Files. The isolates can be found in EnteroBase using the ‘Search Strains’ parameter and under ‘Strain Metadata’, selecting the ‘Name’ option and entering the study sample name in the ‘Value’ box.

The raw genomic sequences are available at NCBI SRA, BioProject ID: PRJNA658685, (SAMN15880274 to SAMN15880361).

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
