# Peer review of "Genomic diversity of Escherichia coli from healthy children in rural Gambia"

_PeerJ, doi:10.7717/peerj.10572_

## Round 0.1 · original submission · Minor Revisions

Even if one reviewer found more problems than the other, I consider those problems to be more of "form" than of essence. Please address all the reviewers concerns.

·

Basic reporting

The study describes the diversity of E. coli collected from healthy children in the rural Gambia between 2007-2011. The authors have shown commensal E. coli differ in their sequence types, phylogenomic groups, virulence, and antibiotic resistance genes. The manuscript is well written. The bioinformatic analysis is excellent, and the data presentation is very well done. Figures and tables are appropriate and informative. A drawback is the low recovery rate of E. coli isolates from frozen stocks.

Experimental design

The authors have carried out an extensive bioinformatic analysis to find the genomic relatedness among the isolates.
Limitation of the study:
E. coli isolates could be recovered only from a total of 21 of the 66 stool samples. Isolates could not be recovered from 9-13 years old frozen stool samples for the remaining 45 samples. Hence, it is not sure if whatever E. coli strains recovered if it reflects true diversity.
Currently, it is not clear if five colonies per sample if sufficient to capture the diversity of E. coli. The recovery rate could influence the analysis of the true diversity of E. coli in the samples. I agree that it is not possible to assure complete diversity even from fresh stool samples. This limitation should be clearly mentioned, discussed, and conclusions (number of STs/individual, immigration, etc.) should be made accordingly.

Validity of the findings

The conclusions should be considered with the limitations due to low recovery issue. The observed diversity could represent only be a fraction of the diverse E. coli population existed.

Reviewer 2 ·

Basic reporting

The manuscript is clearly written in English and easy to read. It cites references appropriately. Figures and tables are appropriate and nicely done.

Experimental design

The research question of this manuscript is clear and well defined. The research seems to be performed properly. Generally speaking methods are properly described however there are a couple of analyses where more detailed methodology is required (see "General comments for the author").

Validity of the findings

Dear Dr. Souza,

I have read in detail the manuscript entitled “Genomic diversity of Escherichia coli from healthy children in rural Gambia” submitted by Foster-Nyarko et al. I enjoyed reading this paper and I find it to be a great contribution to our understanding of the nature and mechanisms driving the genetic diversity of strains isolated from healthy infants from a particular geographical area. I will be happy to recommend this manuscript for publication in PeerJ after my comments are addressed.

Additional comments

Major comments

Abstract:

The abstract suggests that the main goal of this study is to evaluate the mechanisms that generate the within-host diversity of E. coli isolated from healthy children from rural Gambia and it doesn’t mention anything about the accessory gene content findings. I suggest the authors to include some of the virulence and antimicrobial genes and plasmid findings in the abstract otherwise it would be convenient to reconsider the title of the manuscript.

Methods:

I encourage the authors to explain in detail how the migration and within-host evolution events were defined and determined using the SNP data since I find it unclear and it is crucial for this study. I would also like to know more details about how the SNPs where detected, the quality control steps done (including quality score values) and an explanation of why they used different algorithms to do it. Also for accuracy, I request the authors to include the software, their versions and parameters used to process the whole genome raw data and assembly.

I didn’t find reported the access number of the whole genome sequences obtained in this study neither in the main text nor in any table.

Results:

I am confused about how the within-host diversity results section is reported. According to the introduction (Lines 67 to 68) the formation of clouds of intra-clonal diversity refers exclusively to the within-host evolution processes therefore 2 or more within-host evolution events reported in Table 2 are present in just 3 individuals (within-host evolution events column) and not in 13 as it is written in the main text. There is an expanded comment about this below. Please make sure to clarify these results since they are crucial. It might have been a typo. I suggest to report also in the main text the number of immigration events to avoid confusions.

Also, I would have liked to see more details about the “Within-host diversity” section since the authors make special emphasis on the sources of within-host genomic diversity in the abstract, introduction, discussion and conclusions. In addition to other comments (see below) this section would be highly improved if the authors:

- Calculate the nucleotide diversity from the core-genome alignment for all the isolates and by phylogenetic group.
- Estimate if the proportion of immigration events and within-host evolution changes varies per phylogenetic group (discuss these results) and,
- Evaluate the fitness effects of the mutations by obtaining dN/dS ratios.

Discussion:

I would have liked to read about the role of the ecological competition and cooperation among strains (mentioned in the introduction) as well as some of the host traits as factors that determine that different E. coli strains thrive in the hosts.

Also, it would contribute to the discussion if the authors discuss the role of different genetic mechanisms such as point mutation, homologous recombination and, horizontal gene transfer in the generation of the genomic diversity found in their isolates. Although the authors analysed SNPs, it has been demonstrated that homologous recombination is responsible for the generation of a proportion of the core genome diversity (see González-González et al., 2013. Hierarchical clustering of genetic diversity associated to different levels of mutation and recombination in Escherichia coli: A study based on Mexican isolates. Infection, Genetics and Evolution 13: 187-197.)

Supplementary files:

Make sure all the supplementary files are properly cited and numbered in the order they appear in the text.


Minor and specific comments

Introduction:

It would be very good to provide more details about the The Global Enteric Multicenter Study. Although the authors include the references so the reader can go and learn, it would be useful to have a small description about this strain collection to contextualize the results better.

Line 56: I like the ecological mechanisms mentioned and I would like the authors to expand on which epidemiological analyses there are thinking about.


Methods:

Line 86: Explain why ten samples were unavailable for the study.

Line 105: Mention the kit used to prepare the DNA libraries to be sequenced.

Line 107: Although the authors included a list (Supplementary file 5) with the individuals that were sequenced twice, it would be nice to mention the actual number of isolates in the main text.

Line 109: Number supplementary files as they are mentioned in the text.
Supplementary file 5, should be Supplementary file 1 because is the first supplementary file to be mentioned in the text. Also, specify the units of total length (I suppose is “bp”) in Supplementary file 5 Column E.

Line 114: I would like to see a bit more details on this section. I suggest to include the programs (and versions) and parameters used to do the quality check and genome assembly. The authors mentioned that these steps were done as in DeSilva et al., 2016 but I find it more convenient that the reader can obtain this information from the manuscript.

Line 115: Specify if there were any changes in the default when running Snippy.

Line 128: Include MLST in parenthesis after Multi-locus sequence typing.

Line 132: Include cgMLST in parenthesis after core-genome MLST.

Line 158: Include what does cgST stands for in parenthesis.

Line 141: Please include the quality control steps (and the strains used as reference) followed when using CSIPhylogeny tool to detect the SNPs. Also include the read mapping, SNP quality and SNP depth quality scores used.

Line 140: Shouldn’t be “the whole genome sequences” of the strains sequenced twice instead of “duplicate sequence reads”? Duplicate sequence reads normally refer to those raw reads that are sequenced twice by the sequencer and that can be the product of extra PCR cycles during the library prep.

Line 143: Which two steps? Shouldn’t be two different algorithms or software to detect SNPs instead? I suggest that the SNPs detection have its own subsection therefore you can expand more in details about parameters and quality checks used. Also, in this same paragraph the authors can explain in detail how did they differentiated between migration and within-host events. Dixit et al. 2017 describe nicely all this.

Line 151: Explain which software or algorithm was used to calculate the co-occurrence of AMR genes.

Line 164: Do you mean all the pathogenic and commensal E. coli strains from the GEM database, including samples from other countries such as Kenya, India, etc? Clarify and include the number of strains.

Results:

Line 181: I would prefer Table 1 to be a supplementary file.

Line 183: Refer to Supplementary information 2 at the end of “yielding 88 isolates”.

Line 185-186: Besides mentioning that the strains were scattered over all the eight main phylogenetic groups of E. coli, include a more quantitative description about their distribution taking into account all 88 isolates. For example, one way could be to report the proportion or percentage of isolates that belong to phylogenetic group A out of the 88 isolates and so on with the other phylogenetic groups.

Lines 195-197: What does variant refers in this context? What do you mean with SNP cloud? Define. I am confused about how the way these results are reported. According to the introduction (Lines 67 to 68) the formation of clouds of intra-clonal diversity refers exclusively to the within-host evolution processes therefore the within-host evolution events reported in Table 2 are present in just 3 individuals (within-host evolution events column) and not in 13 as it’s written in the text. I am a bit unclear about how the immigration events explain the three quarters of the observed variation calculated according to the information of Table 2. Please make these results as clear as possible. Also report how many SNP differences where detected between the two whole genome assemblies of the same isolate.

Line 201: How did you determine that two or more isolates belonging to the same ST present in the same host are the results of an immigration or within-host evolution event (For example host H-2 in table 2, all 5 isolates belong to the same ST9274)?

Line 208: I suggest moving Table 4 to supplementary information.

Line 211: Could you elaborate more why to focus on ST38? Mention in the main text the message conveyed on each panel of Figure 5.

Line 215: Mention which clinically relevant classes of antibiotics are these.

Line 213: Expand on the results about the encoding putative virulence factors. For example, mention the most prevalent putative virulence factors or you can mention the main categories included in Figure 2 and if they distribute across all the samples, etc. Also test if there is a statistical association of particular virulence factors with ST and phylogenetic group.

Line 220: How large are these plasmids? Include approximated size.

Line 240: Nice! I find this very interesting.

Tables and Figures:

Table 1. Can go to supplementary information

Table 2. I suggest to change “Genotype number” for “Colony or isolate number”. I find colony number more appropriate since for instance, all the colonies isolated from particular individuals belong to the same ST or have the same genotype (Invididual H-2, H-28, etc.).

Figure 5. Panel A. I guess the size of the circle represents the number isolates per clade. Specify this in the figure caption. I assume that persistent strains refer to commensal strains. I suggest to change “persistent” for “commensal” to be consistent with the main text. Make sure that the explanation about the red highlights used to draw attention to a particular isolate in panel C is said for all panels because for example in the figure caption corresponding to Panel A and B, there is no explanation about these red highlights. Also, you say that the geographical locations where isolates were recovered are displayed in Figures 4A-C and Figure 4 correspond to the plasmid content results.

GEMs consent form file. This form is not filled out. I though the authors had obtained the samples from the GEMS collection directly and that they didn’t have to collect them. If this is the case, then, I question the need to include this black consent form.

---

## Round 0.2 · accepted · Accept

Dear authors, this last version was accepted by editor and reviewers, we acknowledge your dedication and care to do all the needed corrections.

Reviewer 2 ·

Basic reporting

The manuscript is clearly written and structured. It also cites relevant references in the field. The figures are self-explanatory and with a nice design.

Experimental design

I feel satisfied of how the authors expanded on how they performed molecular and bioinformatic analysis. Also, the authors present clear aims and their results filled a knowledge gap.

Validity of the findings

Dear Dr. Souza,

I have read in detail the new version of the manuscript entitled “Genomic diversity of Escherichia coli from healthy children in rural Gambia” submitted by Foster-Nyarko et al. and I find satisfactory the way they authors incorporated and addressed my previous comments. I recommend this new version of the manuscript for publication in PeerJ.